# Alu Deletions in *LAMA2* and *CDH4* Genes Are Key Components of Polygenic Predictors of Longevity

**DOI:** 10.3390/ijms232113492

**Published:** 2022-11-04

**Authors:** Vera V. Erdman, Denis D. Karimov, Ilsia A. Tuktarova, Yanina R. Timasheva, Timur R. Nasibullin, Gulnaz F. Korytina

**Affiliations:** 1Institute of Biochemistry and Genetics, Subdivision of the Ufa Federal Research Centre of the Russian Academy of Sciences, 450054 Ufa, Russia; 2Ufa Research Institute of Labor Medicine and Human Ecology, 450106 Ufa, Russia

**Keywords:** longevity, aging, Alu retrotransposon, polygenic predictor, APSampler

## Abstract

Longevity is a unique human phenomenon and a highly stable trait, characterized by polygenicity. The longevity phenotype occurs due to the ability to successfully withstand the age-related genomic instability triggered by Alu elements. The purpose of our cross-sectional study was to evaluate the combined contribution of *ACE*Ya5ACE*, *CDH4*Yb8NBC516*, *COL13A1***Ya5ac1986*, *HECW1*Ya5NBC182*, *LAMA2*Ya5-MLS19*, *PLAT*TPA25*, *PKHD1L1*Yb8AC702*, *SEMA6A*Yb8NBC597*, *STK38L*Ya5ac2145* and *TEAD1*Ya5ac2013* Alu elements to longevity. The study group included 2054 unrelated individuals aged from 18 to 113 years who are ethnic Tatars from Russia. We analyzed the dynamics of the allele and genotype frequencies of the studied Alu polymorphic loci in the age groups of young (18–44 years old), middle-aged (45–59 years old), elderly (60–74 years old), old seniors (75–89 years old) and long-livers (90–113 years old). Most significant changes in allele and genotype frequencies were observed between the long-livers and other groups. The search for polygenic predictors of longevity was performed using the APSampler program. Attaining longevity was associated with the combinations *LAMA2*ID + CDH4*D* (OR = 2.23, P_Bonf_ = 1.90 × 10^−2^) and *CDH4*DD + LAMA2*ID + HECW1*D* (OR = 4.58, P_Bonf_ = 9.00 × 10^−3^) among persons aged between 18 and 89 years, *LAMA2***ID + CDH4*D + SEMA6A*I* for individuals below 75 years of age (OR = 3.13, P_Bonf_ = 2.00 × 10^−2^), *LAMA2*ID + HECW1*I* for elderly people aged 60 and older (OR = 3.13, P_Bonf_ = 2.00 × 10^−2^) and *CDH4*DD + LAMA2*D + HECW1*D* (OR = 4.21, P_Bonf_ = 2.60 × 10^−2^) and *CDH4*DD + LAMA2*D + ACE*I* (OR = 3.68, P_Bonf_ = 1.90 × 10^−2^) among old seniors (75–89 years old). The key elements of combinations associated with longevity were the deletion alleles of *CDH4* and *LAMA2* genes. Our results point to the significance for human longevity of the Alu polymorphic loci in *CDH4*, *LAMA2*, *HECW1*, *SEMA6A* and *ACE* genes, involved in the integration systems.

## 1. Introduction

Longevity is a sociobiological phenomenon of human life, characterized by the ability of individuals to keep their physiological parameters at a qualitatively high level for a long enough period of life and exceeding the life expectancy average for population. The longevity phenotype is obtained under the influence of hereditary, behavioral and environmental factors. Studying the role of heritability for longevity using the twin method shows a high percentage of the genetic factor—from 33% to 48% [1]. To date, a large number of age-associated genetic markers have been identified. This is facilitated by the undying interest in longevity as one of the fundamental problems of humanity, as well as the modern technical capabilities of molecular genetics, which make it possible to conduct the whole-genome and whole-exome sequencing. At the same time, GWAS data demonstrated the functional heterogeneity of age-associated polymorphic genetic variants [2,3,4]. This provides a background for searching the markers of aging and longevity among a variety of signaling pathways within the framework of various hypotheses.

A functional analysis of a number of loci associated with age showed that one of the main causes of aging can be considered the absolute and relative number of lesions in the organism caused by damage in molecules, cells, organs and their systems [5]. In total, these disorders lead to a decrease in functional activity and a violation of homeostasis of the whole organism as well as its parts, without the possibility of complete recovery. Thus, an age-dependent increase in genome instability leads to a limitation in life expectancy and is involved in the manifestation of the frail phenotype. Accordingly, the ability of the system to adjust the optimal balance of the cellular processes to the intrinsic senile background of an aging organism favors the development of a long-lived phenotype.

Mobile genetic elements (MGEs) play a significant role among the endogenous factors of genome instability [6]. Alu elements are one of the most widespread MGE families in the human genome—the number of copies of these elements exceeds 1 million, which is about 11% of the entire DNA [6]. About 30% of all genes include the Alu insertion [7]. A family of Alu repeats belongs to a class of Short Interspersed Nucleotide Elements (SINE), which is included in the subgroup of non-LTR retrotransposons. Structural features and processes of Alu transposition are described in detail in the reviews [6,7,8].

MGE activity is often considered a common occurrence in the genome functioning and in the development of a biological system [9]. The ability of Alu-MGEs to function as enhancers, insulators and alternative splicing sites and participate in other processes in the cell probably explains their wide distribution and indicates the evolutionary advantage of their presence in the human genome [10]. MGE transpositions identified during embryonic development are involved in the regulation of cell and tissue differentiation, and also mediate widespread structural variations in human populations [11]. During the postnatal ontogenesis, activation of certain MGEs in stem cells is necessary for pluripotent maintenance as well as for specific cell differentiation. With age, MGEs can cause various mutations and chromosomal aberrations and affect the epigenetic landscape of the genome, transcription regulation processes and gene expression [7,12]. Thus, by profoundly changing the structure, expression and function of genes, they are involved in the triggering of various pathologies. For example, a recent study has shown the role of Alu in the genes of the angiotensin-converting enzyme (*ACE)* and the tissue-type plasminogen activator (*PLAT)* in susceptibility to infectious agents, such as SARS-CoV-2 [13]. Alu retrotransposons regulate the mechanisms of natural aging by participating in important cellular processes such as proliferation, apoptosis and stress reactions [14]. All this, in general, indicates the role of Alu-MGEs in the adaptation processes that are important for the attainment of the longevity phenotype.

Alu insertions from the young subfamilies Ya5 and Yb8 are the only members of the family that retained transpositional activity and therefore can influence the adaptive abilities of the organism. For our study, we selected the genes with AluYa5 and AluYb8 insertions in their introns, including the genes encoding transcription enhancer factor (*TEAD1*), serin/threonine kinase 38-like (*STK38L*), E3 ubiquitin-protein ligase (*HECW1*), polycystic kidney and hepatic disease 1-like protein 1 (*PKHD1L1*), cell receptor semaforin-6 (*SEMA6A*), R-cadherin adhesion protein (*CDH4*), collagen type 13 alpha 1 chain (*COL13A1*), laminin-2a (*LAMA2*), *PLAT* and *ACE*. Participating in the growth and development, control of metabolism, the immune response and apoptosis and being the blood plasma enzymes, components of extracellular matrix, as well as key transcription factors, these genes are interwoven and important for the physiological and pathological aging processes. Previous reports have linked the *TEAD1*, *HECW1*, *SEMA6A*, *COL13A1* and *LAMA2* genes with different age-related cancers [15,16,17,18,19,20,21], the *PLAT* and *ACE* genes with cardiovascular disease [22,23,24,25], the *ACE* and *HECW1* genes with Alzheimer’s disease [25,26] and the *SEMA6A* gene with type 2 diabetes [27]. The *STK38L* and *PKHD1L1* genes play an important role in the functioning of the immune system [28,29]. The *ACE*, *PLAT* and *CDH4* genes were associated with aging and longevity [20,24,30,31,32,33,34,35].

Many genes exert pleiotropic effects, thus the same allelic variant of a gene can both contribute to longevity and counteract this phenotype, depending on concomitant factors such as the genetic environment. This concept is of particular importance in terms of the role of Alu-MGEs in the end-of-life adaptation, because genetic architecture shaped by natural selection due to the species-specific arrangement of various MGEs relative to each other and protein-coding genes is fundamental for the systemic control of the genome functioning during ontogenesis [7]. Therefore, our research objective was to identify the combined contribution of *ACE*Ya5ACE*, *CDH4*Yb8NBC516*, *COL13A1***Ya5ac1986*, *HECW1*Ya5NBC182*, *LAMA2*Ya5-MLS19*, *PLAT*TPA25*, *PKHD1L1*Yb8AC702*, *SEMA6A*Yb8NBC597*, *STK38L*Ya5ac2145* and *TEAD1*Ya5ac2013* Alu elements to reach the longevity.

## 2. Results

### 2.1. Population Analysis

Our study is the first, to our knowledge, to characterize the age-specific distribution of allele and genotype frequencies of the Alu polymorphic loci in genes associated with aging and longevity among healthy individuals from the Tatar ethnic group, aged between 18 and 113 years old (Table 1). We focused our research on genes encoding blood plasma enzymes (*ACE*, *PLAT*), components of extracellular matrix (*COL13A1*, *LAMA2*), key transcription factors and other functional genes implicated in the aging processes via regulation of homeostasis (*HECW1*, *CDH4*), growth and development (*SEMA6A*, *TEAD1*, *CDH4*), control of metabolism (*HECW1*), immune response (*STK38L*, *PKHD1L1*) and apoptosis (*HECW1*, *STK38L*). All studied loci were tested for compliance with the Hardy–Weinberg equilibrium. In a study involving practically healthy individuals, it is paramount to define the group with the genotype frequency distribution most closely resembling that in the general population without the selection influences. Many common diseases manifest in adulthood, between the ages of 45 and 59 years, most likely affected by the sedentary lifestyle and dietary habits prevalent in the modern world [36]. Another study established 45 years as typical age for the development of a number of common diseases [37]. Accordingly, the group of persons under 45 years was assigned for the population analysis as the most closely resembling the general population. We demonstrated the compliance with the Hardy–Weinberg equilibrium in the young group (18–44 years old): *ACE**Ya5ACE (P_HWE_ = 3.04 × 10^−1^), *HECW1**Ya5NBC182 (P_HWE_ = 2.86 × 10^−1^), *SEMA6A**Yb8NBC597 (P_HWE_ = 2.61 × 10^−1^), *CDH4**Yb8NBC516 (P_HWE_ = 6.60 × 10^−2^), *STK38L**Ya5ac2145 (P_HWE_ = 1.71 × 10^−1^), *PKHD1L1**Yb8AC702 (P_HWE_ = 6.17 × 10^−1^), *TEAD1**Ya5ac2013 (P_HWE_ = 3.12 × 10^−1^), *PLAT**TPA25 (P_HWE_ = 5.10 × 10^−2^), *COL13A1**Ya5ac1986 (P_HWE_ = 1.00), *LAMA2**Ya5-MLS19 (P_HWE_ = 7.25 × 10^−1^).

In the older age groups, among individuals without the chronic disease symptoms, selection may favor polymorphic variants that ensure more stable adaptive phenotypes. This was our main working hypothesis, which provided the basis for the further interrogation of individuals and, to a greater extent, polygenic predictors of longevity.

### 2.2. Age-Dependent Analysis of Individual Alu Polymorphic Loci

One of the most interesting research topics is establishing the possible selection of certain alleles at particular stages of ontogenesis, including the boundaries of the specified age periods. Therefore, we analyzed the age-specific dynamics of the genotype and allele frequencies of the Alu polymorphic loci in ten genes implicated in aging and longevity in our study group of individuals whose age ranged almost over a century—from 18 to 113 years. Pairwise comparison of age groups revealed differences in genotype and allele frequencies of Alu polymorphic loci in eight genes (Appendix A).

For all the Alu polymorphic loci demonstrating these changes, the frequencies of alleles and genotypes begin to fluctuate starting from 60 years old. Statistically significant associations with age for the studied loci are shown in Table 2. In the elderly group we observed the higher frequency of the *HECW1*DD* genotype compared with the young group (17.04% vs. 8.23%, OR = 2.291, P = 4.00 × 10^−3^). The old group demonstrated a two-fold decrease in the *CDH4*DD* genotype frequency compared to the middle-aged group (14.44% vs. 25.25%, OR = 0.5, P = 9.00 × 10^−3^). People older than 90 years had a significant increase in *LAMA2*ID* genotype frequency in comparison with all the other age groups, with the strongest association found when comparing long-livers with the middle-aged individuals (60.26% vs. 41.8%, OR = 2.153, P = 1.49 × 10^−4^, P_Bonf_ = 4.47 × 10^−3^) and the old group (60.26% vs. 44.87%, OR = 1.9, P = 5.51 × 10^−4^, P_Bonf_ = 1.54 × 10^−2^). The *LAMA2*DD* genotype frequency was decreased in the group of long-livers compared to the old group (24.45% vs. 37.21%, OR = 0.546, P = 5.00 × 10^−4^, P_Bonf_ = 1.45 × 10^−2^). Furthermore, the long-livers compared to the old seniors showed significant increase in the *CDH4*DD* genotype frequency (26.22% vs. 14.44%, OR = 2.106, 1.00 × 10^−3^), while the *CDH4*I* allele frequency was decreased (51.83% vs. 63.25%, OR = 0.475, P = 1.00 × 10^−3^, P_Bonf_ = 3.00 × 10^−2^). Comparing the long-livers with the elderly persons, we detected the decrease in the *SEMA6A*D* allele frequency (73.28% vs. 80.78%, OR = 0.471, P = 4.90 × 10^−2^) and even more pronounced decrease in the *HECW1***DD* genotype frequency (9.15% vs. 17.04%, OR = 0.490, P = 2.80 × 10^−2^). The frequency of the *SEMA6A*II* genotype was more than 3-fold higher among long-livers than in the young group (9.77% vs. 3.15%, OR = 3.33, P = 6.00 × 10^−3^).

Considering that sex has been shown to influence the lifespan, we additionally analyzed sex-specific associations between age and Alu polymorphisms in the studied genes (Table 2). The inclusion of sex as a predictor of attaining longevity in the logistic regression model led a two-fold increase in the chances of becoming a long-liver among individuals with the *PKHD1L1*DD* genotype (OR = 2.022, P = 1.20 × 10^−2^) and a mild decrease in the chances of reaching old age among individuals with the *CDH4*D* allele (OR = 0.479, P = 6.00 × 10^−3^). Associations identified for the other studied loci were not changed by the inclusion of sex.

Thus, the group of long-livers demonstrated the most significant changes in genotype and allele frequencies compared to all the other age groups. Moreover, the associations identified in the group of long-livers were the only ones that survived the multiple testing adjustment. Most notably, we established that among the long-livers the frequency of Alu insertion in the Yb8NBC516 locus of the *CDH4* gene was reduced, and the number of *LAMA2*ID* genotype carriers was increased.

### 2.3. Polygenic Analysis of Longevity

Conducting a polygenic analysis required defining the age group that would be used as a control for individuals who had reached longevity. To define this group, we analyzed the results of a single-locus pairwise comparison of allele and genotype frequencies in various age periods and detected the differences, suggesting the possibility that genetic patterns conferring longevity may differ at distinct stages of ontogenesis. This may be due to complex networks of exo- and endogenous factors. During the lifespan, the human organism is affected by a variety of external influences, including those with deferred effects (for example, nutritional status during childhood, radiation, disease burden, psychosocial support, etc.). These factors interact with physiological, hormonal and epigenetic background changes during life. In such conditions, the hereditary background of a person can play different, and sometimes contradictory, combinations encoded by the gene set.

Using the APSampler (Allelic Pattern Sampler) algorithm, we tested all the possible options for our gene set comparing the group of long-livers with each age group, and with their combinations. As a result, we obtained the combinations of genotypes and alleles of the studied Alu polymorphic variants associated with longevity, with the control group established in four variants—18–74 years (1826 combinations), 18–89 years (5109 combinations), 60–89 years (2346 combinations) and 75–89 years (2401 combinations). The combinations with the highest statistical significance after the Bonferroni correction are presented in Table 3. All these polygenic patterns were associated with increased chances of attaining longevity (OR > 2). The Ya5-MLS19 locus of the *LAMA2* gene is the key element that forms all the longevity-associated combinations. Its heterozygous variant in combination with the *Yb8NBC516*D* allele of the *CDH4* gene promotes longevity when compared with all the age groups selected as controls. At the same time, the combination of the *LAMA2*ID* genotype and the *HECW1*I* allele was associated with higher chances of reaching longevity for people over 60 years. When comparing a group of long-livers with persons under 74 years old, the combination of *LAMA2***ID*, *CDH4*D* and *SEMA6A*I* was significantly more frequent (22.22% and 8.37%, respectively, OR = 3.13, P = 1.80 × 10^−5^, P_Bonf_ = 2.00 × 10^−2^). The pattern combining the *CDH4*DD* genotype with *LAMA2***D* and *ACE***I* alleles was more common among long-livers than among persons of the old group (75–89 years) (19.08% vs. 6.02%, OR = 3.68, P = 7.76 × 10^−6^, P_Bonf_ = 1.90 × 10^−2^).

For the elements of the identified combinations, we additionally calculated the individual ORs to compare their individual effects and the effects in combinations (Appendix A). We found that the average individual OR values were around 1.5. We established that the maximum ORs obtained for the *CDH4***DD* (2.11) and *LAMA2***ID* (1.84) loci were still lower than in combinations with the other markers. Thus, using the APSampler algorithm, we successfully identified the combinations of the studied loci significantly associated with longevity.

## 3. Discussion

In our study, we explored the polygenic associations between a complex of Alu polymorphic loci and longevity in the ethnic group of Tatars. The most informative polygenic predictors of longevity included, first of all, Alu deletions in the *LAMA2*, *CDH4* and *HECW1* genes, as well as Alu insertions in the *SEMA6A* and *ACE* genes. Notably, all six most significant patterns of longevity included the Ya5-MLS19 locus of the *LAMA2* gene, and five the Yb8NBC516 locus of the *CDH4* gene. Thus, the genes most prominently associated with human longevity according to our results encode proteins involved in the maintenance of the extracellular matrix and cell adhesion.

Adhesive glycoproteins of the extracellular matrix are represented mainly by large molecules of laminins. They constitute the stroma of the basement membrane and, by binding to cells through affinity receptors, are involved in the organization, attachment and migration of cells in tissues [38]. They also affect the proliferation, differentiation and function of cells connected to basement membranes [39]. Laminins are heterotrimers of α, β and γ chains, each of which have several variants. Combinations of these chains in a polypeptide molecule form at least 15 isoforms of laminin [38]. The α-2 chain, which is one of the subunits of laminin-2 (merosin) and laminin-4 (s-merosin), is encoded by the gene *LAMA2*. The *LAMA2* gene is mainly expressed in basement membranes of striated muscle, Schwann cells and some other tissues [38]. A change in the amino acid composition can cause a violation of the protein structure, which is manifested in the inability to polymerize and form full-fledged basement membranes of myofibrils in muscle tissue [40]. More than 300 mutations in the *LAMA2* gene have been identified so far in patients with congenital myotonic dystrophy type 1 [41]. Laminin-2 is also widely researched in the study of common human diseases at late stages of ontogenesis. A decrease in *LAMA2* gene expression level is observed in some types of cancer and is a predictor of poor survival of cancer patients [15]. Mutations in the *LAMA2* gene also cause various damage to brain structures and changes in the Substantia alba [42], leading to impaired myelination of neurons [43]. These changes are associated with age-related neurodegenerative pathologies. The anti-amyloidogenic role of laminin has been discovered, which is of interest for the development of approaches to the treatment of Alzheimer’s disease [44]. Laminin-2, derived from pericytes, mediates stimulation of oligodendrocyte differentiation after demyelination [45]. The important role of laminin-2 in the organization and maintenance of homeostasis in neurons is also emphasized in the engineering of peripheral nerve repair [46]. Thus, the role of laminins in aging and longevity is studied mainly within the framework of the searching for a connection with age-associated neurodegenerative pathologies. However, for example, an analysis of the laminin levels in the blood serum and cerebrospinal fluid showed no relationship between the concentration of this protein and Alzheimer’s disease and its increase with age [47]. However, a later experiment demonstrated the decrease in the levels of laminin in vascular basement membranes in aged mice relative to young mice [48]. In general, it is noted that the different levels of laminin activity in the studies of aging are explained by tissue specificity [49]. Here, we established a correlation of the heterozygous *LAMA2*Ya5-MLS19* genotype with longevity, that is, the presence of one allele containing the Alu element contributed to an increase in lifespan and may indicate an overdominant effect on the highly adaptive phenotype. In two identified patterns, the presence of the *LAMA2*D* allele provided the long-livers with an advantage over old individuals aged under 90 years. Bearing in mind the inhibitory effect of Alu insertions on the expression of genes containing them [50], it is likely that a moderate and elevated level of laminin-2 is associated with longevity.

Cadherins also belong to the proteins that perform the function of intercellular adhesion and thus ensure the stability of a multicellular organism. Cadherins are cell membrane glycoproteins that carry out calcium-dependent homotypic adhesion during early embryonic development. Cadherins play a key morphogenetic role in tissue development, providing cell–cell interactions necessary for sorting and migration of cells, as well as formation and maintenance of tissue boundaries [51]. In nervous tissue, cadherins are critical for the formation of Nervorum capitalium and the neuronal networks, including the neuronal recognition of effector cells [52]. In a review dedicated to the structure and functioning of the extracellular matrix of the glycocalyx and the blood–brain barrier, particular importance is given to cadherin as an agent that provides adhesion at the pericyte–endothelium interface [49]. More than one hundred members of the cadherin family have been identified, one of which, R-cadherin (cadherin 4, type 1), is encoded by the *CDH4* gene. The R-cadherin gene has been studied mainly in relation to malignancies, and persistent associations of the risk of cancer have been reported with decreased *CDH4* gene expression [16,17]. In contrast to healthy cells with a demethylated promoter, 57–95% of cancer cells have a methylated *CDH4* promoter [18]. An increase in *CDH4* gene expression was revealed in a model of chronic sustained hypoxia in comparison with the control [53]. According to the GWAS results, aging among a Framingham cohort was associated with SNPs rs1970546 and rs2024714 located in *CDH4* genes [18,22]. Another GWAS also revealed the associations of this gene with age-related changes and longevity [33]. Moreover, an analysis of the set of longevity-associated loci demonstrated the enrichment of genes that control the processes of cell adhesion and cell–cell interactions [33]. This suggests the active involvement of genes associated with cancer-related processes (in this case, protection against cancer) in the formation of a stable highly adaptive phenotype contributing to longevity. Regarding the Alu polymorphism of the *CDH4* gene, the presence of the *Yb8NBC516*D* allele predominantly in the homozygous state in combinations associated with longevity may indicate an important role of this gene activity in protecting against cancer.

The combination most significantly associated with longevity included, in addition to the adhesion molecule genes, the Alu deletion allele of the *HECW1* gene. The *HECW1* gene encodes the E3 ubiquitin ligase, which contains the C2 and WW domains and is involved in the ubiquitin-dependent degradation of proteins in proteasomes. This protein is abundant in neuronal tissues such as brain and spinal cord [54]. Thus, due to its participation in protein homeostasis, HECW1 is a key element in the normal and pathological development of nervous system [55]. The *HECW1* gene product controls the cell cycle through the enhancement of the transcriptional and pro-apoptotic activity of p53 [56]. The HECW1 protein also ubiquitinates the mutant form of the superoxide dismutase (SOD1) enzyme in people with hereditary amyotrophic lateral sclerosis, binds to amyloid-sensitive epithelial sodium channels and possibly participates in the regulation of their activity. Another target of HECW1 is the *DVL1* gene, a member of the Wnt pathway, which controls carcinogenesis and embryonic development [54]. Thus, HECW1 is involved in the regulation of morphogenesis, apoptosis and response to oxidative stress. The *HECW1* gene has been linked to oncological disorders [56]. A significant increase in *HECW1* expression was shown in pathological tissues in non-small cell lung cancer [19]. However, *HECW1* was significantly down-regulated in clear cell renal cell carcinoma [20]. In total, the protein degradation controlled by the E3 ubiquitin ligases is surmised to play a fundamental role in the self-renewal, maintenance and differentiation of cancer stem cells [57]. This is essential for the development of a pathological senile phenotype. Moreover, it is interesting that there is an inverse correlation between oncological and neurodegenerative diseases, which can be explained by the multiple roles of the p53 protein signaling network. HECW1 might have opposite effects in tumorigenesis and the development of neuronal diseases by enhancing p53-mediated apoptotic cell death [26]. Thus, in old age, the same triggers can both provoke the diseases and provide resistance to them. The polygenic analysis of associations can assess the accumulation of small independent subthreshold effects of alleles. As a result, complex markers can be qualitatively different from the individual effects of each allele [58]. The Alu insertion allele of the *HECW1* gene can therefore have a different effect on resistance to age-related diseases and the attainment of longevity status. This is demonstrated by our results: the combinations associated with longevity include not only *HECW1*D*, but also the allele containing the Alu insertion.

The Alu insertion allele of the *SEMA6A* gene is another component that modulates the effects of combinations associated with longevity. The protein product of the *SEMA6A* gene is the signaling membrane-bound semaphorin-6. During ontogenesis, the transcriptional activity of the *SEMA6A* gene may change [59]. This may suggest the implication of the *SEMA6A* gene in the development of age-associated phenotypes. Semaphorin-6 is shown to be involved in the structural and functional organization of the nervous system, primarily in axonal guidance [60]. Knockout of the *SEMA6A* gene led to pathologies of the nervous system; in addition, semaphorin-6 plays an important role as a receptor of thalamocortical neurons [61]. Moreover, the *SEMA6A* gene is expressed in certain tumor cells and promotes their growth independent of cell adhesion [62]. SEMA6-induced signals are involved in the regulation of apoptosis, particularly in cancer cells [21]. Various mutations of the *SEMA6A* gene were associated with a range of diseases. For example, rs26595 is associated with autoimmune Wegener’s granulomatosis [63], and rs154576 is associated with a higher risk of Trichophyton tonsurans infection [64]. In addition, impaired expression of the *SEMA6A* gene led to a decrease in sensitivity to the exotoxin TcsL [65], which reveals the participation of semaphorin-6 in resistance to infectious agents. Thus, semaphorin-6 plays an important role in the development of the nervous, cardiovascular and immune systems [66], and can potentially be involved in the pathogenesis of the diseases affecting these systems that mostly develop with age. Semaphorin-6 is essential, particularly, for the development of blood vessels and adult angiogenesis [67]. In type 2 diabetes, inhibition of miR-27a/b, which targets the angiogenesis repressor SEMA6A, promotes better healing of diabetic wounds [27]. The findings on the participation of repressed SEMA6A in regeneration processes are in accordance with our results demonstrating that the presence of the Alu element in the gene that suppresses its activity is associated with longevity.

Among the combinations most significantly associated with longevity according to our results, the one that includes the *Ya5ACE*I* allele of the *ACE* gene encoding angiotensin-converting enzyme occurs three times more frequently in the group of long-livers than in the group of old people (75–89 years). The *ACE***I* allele is associated with a significantly lower level of the vasoconstricting angiotensin II in the blood and, accordingly, with resistance to the development of cardiovascular diseases [23,24]. Previously, this allele has already shown an association with longevity in some human populations [34,35]. However, according to the results of a meta-analysis, the longevity phenotype was associated with the *ACE*D* allele [31]. The authors call this result paradoxical (the genotype and allele associated with a high risk of cardiovascular diseases become more common with age), but they link these findings to the observation that an increase in the angiotensin-converting enzyme level prevents the onset of Alzheimer’s disease [31]. In addition, ACE being an amyloid-degrading enzyme, it can decrease amyloid toxicity [25]. The most recent findings regarding the Alu polymorphic locus in the *ACE* gene are related to the interaction of the human and viral genomes, and consider human longevity as a phenotype resistant to infectious agents. The *ACE*I* allele has shown protective effects against COVID-19 disease and its severe complications [13]. Alu insertions in genes have been demonstrated to increase host resistance to viral infection: double-stranded RNAs transcribed from Alu elements activate antiviral innate immune signaling pathways in mitochondria through MDA5—a viral dsRNA sensor [68]. In total, the favorable role of low ACE activity and the Alu insertion in the *ACE* gene are more pronounced in physiological backgrounds specific to the late stages of ontogenesis.

Our study has several limitations. We focused on the Alu insertion and deletions in ten genes that were chosen based on their potential significance for aging and longevity. Alu elements located elsewhere in the human genome could further elucidate the mechanisms of aging. Additional research of the biological background and functional studies are needed to explain our results. Another limitation of the study concerns the relatively small size of the study group of long-livers, although it should be noted that longevity is the exceptional survival phenotype and therefore is accompanied by certain challenges in finding these people, especially healthy ones, among the population. Further studies with larger and more diverse ethnic populations are required to validate our results. Our study was restricted to the population of Tatars from the Volga-Ural region of Russia, which, in addition to limiting the ability to generalize the results for a broader population, at the same time acts as one of the strengths of our study increasing the ethnic homogeneity of the total sample. That allows us to avoid the problems related to the possible population heterogeneity. To minimize the influence of the external environmental factors that could significantly affect the lifespan and the quality of life, the study sample was comprised of the indigenous population of the Volga-Ural region of Russia, born and permanently residing in the Republic of Bashkortostan. Polygenic analysis was performed using the APSampler algorithm, which does not allow the adjustment for sex, and therefore associations observed for the combinations of the genotypes/alleles of the studied loci could be partly attributed to sex differences in the aging processes. However, this program has proven to be a powerful tool for identifying complex genetic predictors [69].

According to the results of our analysis of genes comprising the polygenic predictors of longevity, the proteins maintaining the integrity and composition of complex structures of the organism are especially important for preserving the homeostasis and success of the later stages of ontogenesis. Moreover, all these elements are involved in the structural and functional organization of the nervous system, both under normal conditions and at exposure to endo- and exogenous destabilizing factors (viruses, stress, inflammaging). Age-dependent DNA destabilization leads to the activation of retrotransposons, including Alu, and initiates the neurodegenerative processes [70]. This is very important, because in addition to the increasing with age risk of cancer and cardiovascular disorders, the conditions of modern society provoke the development of neuropsychiatric disorders. The necessity to process the rapidly growing volume of digital data affects the functioning of the nervous system, with its evolutionarily conserved mechanisms for processing information. Alu insertions can provide a molecular background for increasing plasticity and adaptation to certain extreme conditions for the nervous system. For other structures of an organism, stability can be important and advantageous at all stages of ontogenesis, and therefore selection can favor Alu deletions. By affecting gene activity, Alu deletions and Alu insertions together can modulate the effects that influence the development of a highly adaptive phenotype of longevity.

## 4. Materials and Methods

### 4.1. Study Group

Our study was conducted according to the ethical principles for biomedical research involving human subjects outlined in the Declaration of Helsinki of the World Medical Association in 2013, and with the approval of the Bioethics Committee of the Institute of Biochemistry and Genetics of Ufa Federal Research Center of Russian Academy of Sciences (protocol No. 1 dated 28 October 2007). A total of 2054 individuals aged 18 to 113 were randomly selected among the residents of the Republic of Bashkortostan, located in Volga-Ural region of Russia. The groups of elderly people were collected in the City Boarding Homes for the Elderly and Disabled of Ufa and Oktyabrsky, the Ufa War Veterans Hospital and the Central District Hospitals of the Republic of Bashkortostan. The groups of young and middle-aged individuals were selected among those who attended the Ufa City Hospitals and the Central District Hospitals of the Republic of Bashkortostan for regular medical checkup. All volunteers signed the informed consent to participation in the study. DNA samples used in the study were anonymized. To avoid potential risk of distorting results arising from population stratification our sample was ethnically homogenous. All study participants identified themselves and their ancestors in three generations as ethnic Tatars. The total sample was divided into age groups of the young (18–44 years old), middle-aged (45–59 years old), elderly (60–74 years old), old seniors (75–89 years old) and long-livers (over 90 years old) according with the WHO recommendations [71]. The characteristics of the differentiated age groups are presented in Table 4. Total group included healthy individuals without disorders of cardiovascular and nervous systems. Among elderly and old people, as well as long-livers, for whom age-related functional changes in the cardiovascular system, with rare exceptions, are practically the norm, a history of atherosclerosis, ischemic heart disease and cerebral sclerosis was allowed. We established a special criterion of vitality for the long-livers, which we defined as the ability to take care of themselves, maintain physical activity and the preservation of lucidity.

### 4.2. DNA Collection

The samples of 8 μL peripheral venous blood were collected using a vacuum system and stored at −4 °C. DNA was isolated from blood using a standard phenol-chloroform extraction method. DNA samples were stored in 96% ethanol. For PCR procedures, aliquots of DNA samples were dried, dissolved in deionized water and their concentration was equalized (50 ng/μL). The DNA quality was assessed by electrophoresis in 0.8% agarose gel and quantified by ultraviolet absorbance spectrophotometric analysis. DNA aliquot solutions were stored at −20°.

### 4.3. Genotyping

Information about Alu polymorphic regions of studied genes was collected from genomic databases TranspoGene, RepeatMasker and dbRIP. Localization of insertions within genes, their genomic context and flanking sequence data were obtained using the UCSC genomic browser (http://genome.ucsc.edu, accessed on 15 April 2019). Oligonucleotide primers were designed using the Primer-BLAST software (https://blast.ncbi.nlm.nih.gov/Blast.cgi, accessed on 25 April 2019). The insertion or deletion of the Alu element in the studied locus was determined by PCR procedure followed by separation of fragments of the amplified DNA regions using electrophoresis in 2% agarose gel. The list of Alu polymorphic loci and conditions of PCR procedure are presented in Table 5.

### 4.4. Statistical Analysis

Statistical processing of genotyping results was carried out using the software GenePop (v.3.1, Montpellier, France), APSampler (v.3.6.1, San Diego, CA, USA) and the statistical software package SPSS (v.21.0, Chicago, IL, USA). The match of the observed distribution of genotype frequencies to the theoretically expected Hardy–Weinberg equilibrium was assessed by the chi-square test. Genotype and allele frequencies in age groups were compared in pairs using Fisher’s exact two-tailed test. The estimation of associations between the studied Alu polymorphic loci and age was performed using logistic regression analysis. The search for combinations of the studied Alu polymorphic markers associated with longevity was carried out using the APSampler; the software uses the Markov chain Monte Carlo method based on Bayesian approaches [58] and is available at http://apsampler.sourceforge.net/, accessed on 7 May 2021. Taking into account the multiple comparisons, the Bonferroni correction was used, while the differences were considered significant at P_Bonf_ < 0.05.

## 5. Conclusions

For the first time, we analyzed the dynamics of the allele and genotype frequencies of the Alu polymorphic loci in genes significant for longevity among healthy people aged between 18 and 113 years old in the ethnically homogeneous population of the Volga-Ural region of Russia. Significant changes in allele and genotype frequencies were observed between the long-livers and other groups. The key elements of polygenic predictors of longevity were the Alu deletion alleles of the *CDH4* and *LAMA2* genes, which control the physiological functioning of the system maintaining cell interactions. The presence of the Alu deletion and Alu insertion alleles of the *HECW1* gene in combinations associated with longevity may be mediated by dual effects resulting in age-related diseases. The modulating components of polygenic models of longevity are Alu insertion alleles of the *SEMA6A* and *ACE* genes involved in resistance to infectious agents and associated with tolerance to multifactorial age-related diseases. All five elements of the identified polygenic predictors of longevity (Alu polymorphic loci of *CDH4*, *LAMA2*, *HECW1*, *SEMA6A* and *ACE* genes) are involved in the development and stability of structures of the nervous system, and in general in the maintaining of integration systems of the body. It points to the significance of the identified polygenic patterns for longevity.

## Figures and Tables

**Table 1 ijms-23-13492-t001:** Genotype frequencies (%) of the studied Alu polymorphic loci in age-divided groups.

Genotype/ Allele	Young (18–44 Years Old)	Middle-Aged (45–59 Years Old)	Elderly (60–74 Years Old)	Old Seniors (75–89 Years Old)	Long-Livers (90–113 Years Old)	P_HWE_^Y^
*ACE* Ya5ACE						
DD	28.95	27.31	29.11	29.52	25.34	3.04 × 10^−1^
ID	47.72	54.63	47.15	45.93	51.13	
II	23.32	18.06	23.73	24.55	23.53	
D	52.82	54.63	52.69	52.48	50.9	
I	47.18	45.37	47.31	47.52	49.1	
*HECW1* Ya5NBC182						
DD	8.23	16.79	17.04	12.30	9.15	2.86 × 10^−1^
ID	46.75	39.69	42.59	41.73	46.41	
II	45.02	43.51	40.37	45.97	44.44	
D	31.60	36.64	38.33	33.17	32.35	
I	68.40	63.36	61.67	66.83	67.65	
*SEMA6A* Yb8NBC597						
DD	61.02	66.91	66.42	62.43	56.32	2.61 × 10^−1^
ID	35.83	28.06	28.73	31.98	33.91	
II	3.15	5.04	4.85	5.59	9.77	
D	78.94	80.94	80.78	78.42	73.28	
I	21.06	19.06	19.22	21.58	26.72	
*CDH4* Yb8NBC516						
DD	17.80	25.25	18.98	14.44	26.22	6.60 × 10^−2^
ID	40.84	41.41	41.20	44.61	43.9	
II	41.36	33.33	39.81	40.95	29.88	
D	38.22	45.96	39.58	36.75	48.17	
I	61.78	54.04	60.42	63.25	51.83	
*STK38L* Ya5ac2145						
DD	80.72	82.55	81.65	79.96	76.97	1.71 × 10^−1^
ID	17.27	15.44	15.73	18.34	21.91	
II	2.01	2.01	2.62	1.70	1.12	
D	89.36	90.27	89.51	89.13	87.92	
I	10.64	9.73	10.49	10.87	12.08	
*PKHD1L1* Yb8AC702						
DD	22.35	20.00	19.21	27.50	30.2	6.17 × 10^−1^
ID	51.76	58.13	60.93	52.17	49.5	
II	25.88	21.88	19.87	20.33	20.3	
D	48.24	49.06	49.67	53.58	54.95	
I	51.76	50.94	50.33	46.42	45.05	
*TEAD1* Ya5ac2013						
DD	25.50	28.38	26.09	27.45	23.63	3.12 × 10^−1^
ID	46.61	42.57	47.83	44.72	45.6	
II	27.89	29.05	26.09	27.83	30.77	
D	48.80	49.66	50.00	49.81	46.43	
I	51.20	50.34	50.00	50.19	53.57	
*PLAT* TPA25						
DD	31.70	32.28	29.05	30.76	33.33	5.10 × 10^−2^
ID	44.12	50.26	43.24	46.71	40.38	
II	24.18	17.46	27.70	22.53	26.29	
D	53.76	57.41	50.68	54.11	53.52	
I	46.24	42.59	49.32	45.89	46.48	
*COL13A1* Ya5ac1986						
DD	5.63	10.06	7.77	7.06	9.42	1.00
ID	36.62	34.32	42.07	37.27	32.74	
II	57.75	55.62	50.16	55.67	57.85	
D	23.94	27.22	28.80	25.69	25.78	
I	76.06	72.78	71.20	74.31	74.22	
*LAMA2* Ya5-MLS19						
DD	29.59	36.51	33.23	37.21	24.45	7.25 × 10^−1^
ID	48.64	41.80	47.28	44.87	60.26	
II	21.77	21.69	19.49	17.92	15.28	
D	53.91	57.41	56.87	59.65	54.59	
I	46.09	42.59	43.13	40.35	45.41	

Abbreviations: D—Alu deletion allele; I—Alu insertion allele; P_HWE_^Y^—*p*-value of the Hardy–Weinberg equilibrium for the young group.

**Table 2 ijms-23-13492-t002:** Associations of Alu polymorphic loci with age.

Gene Alu Element	Genotype/ Allele	Reference Group	OR (CI_OR_)	P	Sex-Adjusted OR (CI_OR_)	Sex-Adjusted P
Long-livers						
*LAMA2* Ya5-MLS19	DD	Middle-aged	0.563 (0.369–0.859)	8.00 × 10^−3^	0.568 (0.358–0.903)	1.70 × 10^−2^
		Elderly	0.651 (0.444–0.953)	2.70 × 10^−2^	0.648 (0.440–0.956)	2.90 × 10^−2^
		Old seniors	0.546 (0.389–0.768)	5.00 × 10^−4^ *	0.554 (0.391–0.785)	1.00 × 10^−3^
	ID	Young	1.633 (1.145–2.328)	7.00 × 10^−3^	1.491 (0.933–2.393)	9.50 × 10^−2^
		Middle-aged	**2.153** (1.449–3.201)	1.49 × 10^−4^ *	1.914 (1.237–2.959)	4.00 × 10^−3^
		Elderly	1.724 (1.214–2.448)	2.00 × 10^−3^	1.726 (1.208–2.466)	3.00 × 10^−3^
		Old seniors	1.900 (1.391–2.596)	5.51 × 10^−4^ *	1.815 (1.318–2.501)	2.66 × 10^−4^
*CDH4* Yb8NBC516	II	Young	0.604 (0.389–0.939)	2.50 × 10^−2^	0.612 (0.345–1.083)	9.2 × 10^−2^
		Old seniors	0.614 (0.419–0.900)	1.20 × 10^−2^	0.670 (0.453–0.989)	4.40 × 10^−2^
	I	Young	0.609 (0.366–1.013)	5.60 × 10^−2^	0.680 (0.351–1.315)	2.50 × 10^−1^
		Elderly	0.659 (0.405–1.072)	9.30 × 10^−2^	0.659 (0.402–1.080)	9.80 × 10^−2^
		Old seniors	**0.475** (0.308–0.733)	1.00 × 10^−3^ *	0.514 (0.330–0.800)	3.00 × 10^−3^
	DD	Old seniors	**2.106** (1.365–3.249)	1.00 × 10^−3^	1.946 (1.250–3.029)	3.00 × 10^−3^
*SEMA6A* Yb8NBC597	DD	Elderly	0.652 (0.440–0.965)	3.30 × 10^−2^	0.640 (0.430–0.955)	2.90 × 10^−2^
	D	Elderly	**0.471** (0.223–0.996)	4.90 × 10^−2^	**0.487** (0.228–1.042)	6.40 × 10^−2^
	II	Young	**3.330** (1.404–7.899)	6.00 × 10^−3^	**3.794** (1.257–11.449)	1.80 × 10^−2^
*PKHD1L1* Yb8AC702	DD	Middle-aged	1.730 (1.060–2.825)	2.80 × 10^−2^	**2.022** (1.168–3.500)	1.20 × 10^−2^
		Elderly	1.820 (1.202–2.756)	5.00 × 10^−3^	1.780 (1.166–2.718)	8.00 × 10^−3^
	ID	Elderly	0.629 (0.439–0.901)	1.10 × 10^−2^	0.651 (0.451–0.940)	2.20 × 10^−2^
*PLAT* TPA25	II	Middle-aged	1.686 (1.039–2.735)	3.40 × 10^−2^	1.634 (0.953–2.803)	7.40 × 10^−2^
*COL13A1* Ya5ac1986	ID	Elderly	0.670 (0.468–0.960)	2.90 × 10^−2^	0.672 (0.466–0.968)	3.30 × 10^−2^
*HECW1* Ya5NBC182	DD	Elderly	**0.490** (0.260–0.925)	2.80 × 10^−2^	**0.473** (0.249–0.903)	2.30 × 10^−2^
Old seniors						
*ACE* Ya5ACE	ID	Middle-aged	0.706 (0.518–0.960)	2.70 × 10^−2^	0.706 (0.518–0.962)	2.70 × 10^−2^
*LAMA2* Ya5-MLS19	DD	Young	1.410 (1.048–1.897)	2.30 × 10^−2^	1.368 (0.994–1.882)	5.50 × 10^−2^
*CDH4* Yb8NBC516	DD	Middle-aged	0.500 (0.296–0.842)	9.00 × 10^−3^	**0.479** (0.283–0.812)	6.00 × 10^−3^
*PKHD1L1* Yb8AC702	DD	Elderly	1.596 (1.138–2.237)	7.00 × 10^−3^	1.591 (1.134–2.231)	7.00 × 10^−3^
	ID	Elderly	0.699 (0.528–0.927)	1.30 × 10^−2^	0.708 (0.534–0.939)	1.70 × 10^−2^
Elderly						
*HECW1* Ya5NBC182	DD	Young	**2.291 (1.300–4.038)**	4.00 × 10^−3^	**2.459 (1.306–4.631)**	5.00 × 10^−3^
*PKHD1L1* Yb8AC702	ID	Young	1.453 (1.037–2.036)	3.00 × 10^−2^	1.380 (0.929–2.050)	1.10 × 10^−1^
*PLAT* TPA25	II	Middle-aged	1.811 (1.151–2.851)	1.00 × 10^−2^	1.747 (1.095–2.787)	1.90 × 10^−2^

Abbreviations: D—Alu deletion allele; I—Alu insertion allele; OR—odds ratio; CI_OR_—95% confidence interval for OR; P—*p*-value; * indicates P-value that retains statistical significance after adjusting for multiple testing using Bonferroni correction (exact P_Bonf_ are given in the text); associations with OR < 0.5 and OR > 2 are shown in bold.

**Table 3 ijms-23-13492-t003:** Allelic/genotype combinations most significantly associated with longevity according to APSampler analysis.

Combinations	Compared Age Periods ^#^					P	P_Bonf_	OR	CI_OR_
	18–74	18–89	60–89	75–89	90–113				
*LAMA2 Ya5-MLS19*ID + CDH4 Yb8NBC516*DD + HECW1 Ya5NBC182*D*		3.77			15.22	1.70 × 10^−6^	9.00 × 10^−3^	4.58	2.56–8.21
*LAMA2 Ya5-MLS19*D + CDH4 Yb8NBC516*DD + HECW1 Ya5NBC182*D*				4.76	17.39	1.09 × 10^−5^	2.60 × 10^−2^	4.21	2.23–7.96
*LAMA2 Ya5-MLS19*D + CDH4 Yb8NBC516*DD + ACE Ya5ACE*I*				6.02	19.08	7.76 × 10^−6^	1.90 × 10^−2^	3.68	2.09–6.49
*LAMA2 Ya5-MLS19*ID + CDH4 Yb8NBC516*D + SEMA6A Yb8NBC597*I*	8.37				22.22	1.08 × 10^−5^	2.00 × 10^−2^	3.13	1.90–5.16
*LAMA2 Ya5-MLS19*ID + CDH4 Yb8NBC516*D*		28.12			46.58	3.76 × 10^−6^	1.90 × 10^−2^	2.23	1.59–3.13
*LAMA2 Ya5-MLS19*ID + HECW1 Ya5NBC182*I*			40.75		60.27	8.29 × 10^−6^	1.90 × 10^−2^	2.21	1.55–3.15

Abbreviations: ^#^—the data for the compared age groups are presented as the frequencies of allelic/genotype combinations (%); D—Alu deletion allele; I—Alu insertion allele; the asterisk is used to separate the name of polymorphism and allele or genotype; P—significance level estimated using Fisher’s exact test; P_Bonf_—significance level corrected for multiple testing using Bonferroni correction; OR—odds ratio; CI_OR_—95% confidence interval for OR.

**Table 4 ijms-23-13492-t004:** Characteristic of studied group.

Age Group	Age Range, Years Old	Sample Size, *n*	Mean Age ± SD, Years Old	Male/Female, *n* (%)
Young	18–44	542	31.92 ± 7.82	390/152 (71.96/28.04)
Middle-aged	45–59	261	50.87 ± 4.42	152/109 (58.04/41.76)
Elderly	60–74	321	68.39 ± 3.93	107/214 (33.33/66.67)
Old seniors	75–89	693	80.48 ± 3.74	287/406 (41.41/58.59)
Long-livers	90–113	237	93.16 ± 2.97	39/198 (16.46/83.54)
Total	18–113	2054	63.48 ± 22.65	975/1079 (47.47/52.53)

**Table 5 ijms-23-13492-t005:** Nomenclature, genetic localization, amplification conditions and length of the amplificated alleles for the studied Alu polymorphic loci.

Alu Element	Gene, Chromosome Location	Primers	Annealing Temperature (°C)	Alleles (Fragment Length, bp)
Ya5ACE	*ACE* 17q23.3	F 5’-ctg gag acc act ccc atc ctt tct-3’ R 5’-gat gtg gcc atc aca ttc gtc aga t-3’	68	I (490) D (190)
Ya5NBC182	*HECW1* 7p13	F 5′-gaa gga cta tgt agt tgc aga agc-3′ R 5′-aac cca gtg gaa aca gaa gat g-3′	64	I (563) D (287)
Yb8NBC597	*SEMA6A* 5q23.1	F 5′-tga ggt gtt gca gac gat gt-3′ R 5′-cgc atg ctt tag aga ata ccc-3′	63	I (429) D (108)
Yb8NBC516	*CDH4* 20q13.33	F 5′-ggg ctc agg gat act atg ctc-3′ R 5′-gcc tag gcc tac cac tca ga-3′	60	I (445) D (124)
Ya5ac2145	*STK38L* 12p11.23	F 5′-tgt tct aat gac cat gcc tac tt-3′ R 5′-tgc ctt tag gaa gct aca gat tta-3′	60	I (465) D (135)
Yb8AC702	*PKHD1L1* 8q23.2	F 5′-tgt ttg gaa ata agc caa aca at-3′ R 5′-ggg tag caa cct ttt tca tct tt-3′	60	I (482) D (161)
Ya5ac2013	*TEAD1* 11p15.2	F 5′-tgg cag att ctg act ggc ta-3′ R 5′-cac gta agg tga aaa ggg ga-3′	60	I (489) D (212)
TPA25	*PLAT* 8p11.21	F 5′-caa cca atg aaa acc act ga-3′ R 5′-gtt ctc ctg aca tct tta ttg-3′	60	I (518) D (217)
Ya5ac1986	*COL13A1* 10q22.1	F 5′-tct agt ggg atg agg ata ac-3′ R 5′-tgt gcc atg ggg taa gaa ac-3′	60	I (431) D (134)
Ya5-MLS19	*LAMA2* 6q22.33	F 5′-cta tga cgg agt aaa aag aag t-3′ R 5′-gaa aga gtg cca acc ctg tcc-3′	63 (7 cycles) 60 (22 cycles)	I (401) D (106)

Abbreviations: F—forward primer; R—reverse primer; bp—base pair; D—Alu deletion allele; I—Alu insertion allele.

## Data Availability

Data will be available upon request.

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
