# Peer review of "Alu Deletions in LAMA2 and CDH4 Genes Are Key Components of Polygenic Predictors of Longevity"

_ijms, 2022, doi:10.3390/ijms232113492_

Round 1
Reviewer 1 Report
In this study, the authors examined the associations of Alu mobile elements individually, and in combinations, with age and longevity. This study focuses on a unique population of ethnic Tatar in Russia. The authors gathered a relatively large sample of 2054 subjects aged 18 to 113 years and split this sample into five groups of young, middle-aged, elderly, old seniors, and long-living populations. The authors’ goal was to identify the combined contribution of Alu elements harbored by ten genes associated with aging and longevity in previous studies.
In general, this is an interesting paper focusing on the important problem of identifying genetic contributions to survival at extreme ages. The main strength of this study is the focus on a unique population of ethnic Tatar. The authors emphasized the effects of combinations of genetic variants and age groups on the chances to survive 90 years and older. They reported significant associations of six such combinations after Bonferroni correction for multiple testing.
The downside of this work is the lack of validation of the main findings. Furthermore, it is unclear if such validation is feasible given the uniqueness of this population and potential problems with increasing the sample size. Validation of these findings is important because it is unclear whether the identified combinations of genetic factors and age groups are specific for this particular sample or they can be generalized to a broader population of ethnic Tatar. At this stage, this paper looks a discovery report of findings, which could be also false positives due to the sampling procedure. Unless the authors will validate their findings, they should position their work as a potential discovery, which requires validation. Although the authors thoroughly discussed the biology behind their findings, this biological analysis does not substitute validation of their findings. Potentially, the results could be strengthened by performing sensitivity analysis by varying cut-off for the longevity group (defined as 90 years and older).
The following comments may be helpful to improve this manuscript.
1. P-values should be given uniformly across the paper using, for example, scientific notations with two decimals.
2. The authors should indicate how many tests they count as independent to adjust their analyses for multiple tests.
3. Lines 173-176. This sentence is not clear.
4. The analyses without adjustment for sex look problematic as the proportion of males and females substantially varies across the selected age groups.
5. It is unclear whether the results of the sex-adjusted analyses in Table 2 attained the Bonferroni-adjusted significance.
6. Line 182: I could not see the results for LAMA2*ID in Table 2.
7. Table 3: Please indicate that you report frequencies of variants in the “Comparable age periods” columns.
8. Lines 415-416. This sentence looks like a fragment.
Reviewer 2 Report
This study investigated the relationship between Alu-elements in age-related genes and longevity in a homogeneous ethnic population. This study was novel and of interest to the aging research community. There were some grammatical and syntax errors that could be improved. In addition,
#1 The sentence that begins “The Alu family was originated…” is confusing and may not be necessary as background information for this study.
#2 In the results section, when introducing the genes of interest, a few sentences explaining how these individual genes are implicated in aging should be added.
#3 In Table 1, the PHWE for two of the chosen genes, CDH4 Yb8NBC516 and PLAT TPA25, are close to significance suggesting that they do not follow HWE. Does the non-HWE distribution of these genes in the population influence the statistics?
#4 In section 2.2, in the sentence that beings “Furthermore, the long-livers compared to the persons of old age…”, the term “persons of old age” does not correspond to a designated age group in this study. Please clarify exactly what age group is being compared to the long-livers.
#5 In section 2.3 and in Table 3, why were the age groups change from the groups previously used? Also, why do these new age groups overlap?
#6 The gene abbreviations are defined in the beginning of the discussion. However, these genes should be defined in the beginning of the manuscript at the first mention of them.
#7 Since CDH4 had a PHWE of 0.06 and is not necessarily distributed according to HWE, does that affect the OR and does that affect that fact that CDH4 was identified as one of the main contributors to longevity. In other words, is CDH4 only identified to contribute to longevity because of an artifact of an almost significant PHWE?
#8 In the discussion, it was mentioned that sex plays a role in aging where females are more likely to become long-livers. However, there was an unequal number of males and females in each age group, especially in the young and long-livers. Does this difference skew the results? Is there a way to keep the ratio of males/females the same across groups?
#9 It appears that some editing notes were left in the final draft of the manuscript. The last sentence of the first paragraph of section 2.1, the last three sentences of section 4.2, and the last sentence of section 4.3 should probably not be included in the final draft.
Round 2
Reviewer 1 Report
I still disagree with the authors about the models with no adjustment for sex because the results with adjustment for sex are quite different. This difference indicates that sex partly explains the associations observed in the model without adjustment for sex. However, this is up to the authors if they want to present the results without adjustment for sex.
Also, in line 171 it is better to write "compared to" rather than "in compare with".
Author Response
Dear reviewer,
Thank you for your opinion, we agree that this is an important issue, and we revised the Limitations section to acknowledge it.
Thank you again for your careful review, all your comments are very valuable and helped to improve the manuscript.